# Parameters of Hemostasis in Sheep Implanted with Composite Scaffold Settled by Stimulated Mesenchymal Stem Cells—Evaluation of the Animal Model

**DOI:** 10.3390/ma14226934

**Published:** 2021-11-16

**Authors:** Aleksandra Pliszczak-Król, Zdzisław Kiełbowicz, Jarosław Król, Agnieszka Antończyk, Marianna Gemra, Piotr Skrzypczak, Przemysław Prządka, Dariusz Zalewski, Janusz Bieżyński, Jakub Nicpoń

**Affiliations:** 1Department of Immunology, Pathophysiology and Veterinary Preventive Medicine, Wroclaw University of Environmental and Life Sciences, 50-375 Wroclaw, Poland; marianna.gemra@upwr.edu.pl; 2Department and Clinic of Surgery, Wroclaw University of Environmental and Life Sciences, 50-375 Wroclaw, Poland; zdzislaw.kielbowicz@upwr.edu.pl (Z.K.); agnieszka.antonczyk@upwr.edu.pl (A.A.); piotr.skrzypczak@upwr.edu.pl (P.S.); przemyslaw.przadka@upwr.edu.pl (P.P.); janusz.biezynski@upwr.edu.pl (J.B.); jakub.nicpon@upwr.edu.pl (J.N.); 3Department of Pathology, Wroclaw University of Environmental and Life Sciences, 50-375 Wroclaw, Poland; jaroslaw.krol@upwr.edu.pl; 4Department of Genetics, Plant Breeding and Seed Production, Wroclaw University of Environmental and Life Sciences, 50-375 Wroclaw, Poland; dariusz.zalewski@upwr.edu.pl

**Keywords:** hemostasis, coagulation, fibrinolysis, scaffold implantation, sheep

## Abstract

Implantation of composite scaffolds could be potentially associated with the risk of hemostatic disturbances in a recipient. However, there is a lack of information on possible alterations in clotting mechanisms resulting from such a procedure. The aim of the present work was to investigate changes in hemostatic parameters in sheep implanted with a scaffold composed of poly(ε-caprolactone) and hydroxyapatite and tricalcium phosphate (9:4.5:4.5), settled previously with mesenchymal stem cells stimulated by fibroblast growth factor-2 and bone morphogenetic protein-2. Nine Merino sheep were examined for 7 days, and measurements of clotting times (PT, aPTT), activities of antithrombin, protein C and clotting factors II-XII, and concentrations of fibrinogen and D-dimer were carried out before and 1 h, 24 h, 3 days and 7 days after scaffold implantation. The introduction of scaffold initially resulted in a slowdown of the clotting processes (most evident 24 h after surgery); PT and aPTT increased to 14.8 s and 33.9 s, respectively. From the third day onwards, most of these alterations began to return to normal values. The concentration of fibrinogen rose throughout the observation period (up to 8.4 g/L), mirroring the ongoing inflammatory reaction. However, no signals of significant disturbances in hemostatic processes were detected in the sheep tested.

## 1. Introduction

The loss of bones and soft tissues, as well as degenerative processes and necrotic lesions in the facial area, as the result of mechanical injuries, congenital abnormalities or chronic disorders constitute a serious problem in reconstructive cranial surgery in humans [1,2]. Destruction of the local blood vessels and nerves, high probability of infections with oral or nasal bacteria, patient discomfort from difficulty swallowing and negative aesthetic effects make this issue more complex and challenging for researchers and practitioners [1,3,4]. Owing to the complexity of pathological processes associated with facial trauma, there is a need to determine the best methods of therapy, and in particular, methods of removal and reconstruction of the destroyed tissues, stimulation of healing processes and restoration of normal function of the affected area. This is usually accomplished by various grafting procedures, such as autograft or allograft. These methods, however, have some limitations and cause a lot of inconveniences for both donor and recipient [4,5,6]. The size of the material that can be collected for transplantation is often limited, there is a risk of graft rejection and the procedure itself is not infrequently accompanied by infection, bleeding, edema or pain. A useful alternative for auto- or allografts is the implantation of a substitute for the damaged tissue, i.e., a scaffold. Scaffolds are three-dimensional porous structures made of various materials (ceramic materials, synthetic polymers, natural polymers) [3,5,7]. These materials can also be composites, combining ceramic components with various polymers. Scaffolds should have very high biocompatibility, biodegradability and a sufficient mechanical strength. The architecture and properties of scaffolds ought to promote healing and regeneration of the destroyed tissues—they need to be not only a supporting frame but also provide a suitable environment for settlement, multiplication and differentiation of progenitor cells, as well as for the formation of an extracellular matrix, blood vessels and nerves [1,3,5,7]. Before implantation, newer-generation scaffolds are often colonized or impregnated with appropriate cells or specific factors (e.g., stem cells, growth factors, drugs or genes [1,3,7,8]. Prepared in such a way, they not only fill even ample tissue loss but also integrate themselves with the surrounding tissues, triggering their reactiveness and positively affecting healing processes [7,8]. Reparation and regeneration of the lost tissues are then more effective, owing to a better mineralization of the bone and a reduction in ectopic growth, more efficient oxygen and nutrient supply, as well as a decrease in the accompanying inflammatory reaction and clotting processes, making the time of overall healing noticeably shorter [7].

The development of new regenerative techniques and biomaterials makes an animal model necessary for studying interactions between implant and host tissues, including inflammatory and hemostatic reactions, as well potential cytotoxicity of the biomaterial [9]. In recent years, sheep have become a popular experimental model for orthopaedic research and for the development of dental implant systems [10,11]. Sheep, having a body weight similar to that of humans, also display a similar bone-healing rate and bone remodelling activity. Moreover, it has been noticed that sheep and humans have a comparable rate of bone ingrowth into porous implants [9].

The insertion of an implant into a body is always accompanied by inflammatory and hemostatic processes, the severity of which can vary depending on the extent of tissue damage, scaffold preliminary treatment and chemical composition of the biomaterial used. This makes healing of post-implant wounds a very complex process, with final results, time of duration and possible complications determined not only by mutual cellular interrelationships, microenvironment and molecular signals but also by interactions between particular components of hemostasis [12,13,14,15,16,17,18]. Although there is abundant literature data on clotting and fibrinolysis phenomena in bone and soft-tissue engineering, much of this research focuses on effects of local hemostatic processes, and only a few reports are devoted to the whole-body reaction during the healing process following scaffold implantation [18,19,20]. In particular, information on hemostatic phenomena in sheep subject to surgical insertion of various biomaterials is very scarce. Such research would be useful not only for the understanding of the underlying mechanisms of the scaffold, such as host interaction in the animal experimental model, but also for the prediction of clotting reactions in human recipients of that scaffold. According to the data available, coagulation processes in sheep are quite similar to those in humans with regard to some measurable parameters and the overall dynamic of clot formation [21,22,23].

Considering the above, the aim of the present study was to evaluate selected plasma hemostatic parameters in the first week after implantation of a composite scaffold settled with bone marrow mesenchymal stem cells (BM-MSCs) stimulated by fibroblast growth factor-2 (FGF-2) and bone morphogenetic protein-2 (BMP-2) into the ovine mandible. In particular, the following parameters were determined: prothrombin time (PT), activated partial thromboplastin time (aPTT), fibrinogen concentration (Fb), stabilized fibrin degradation product (D-dimer), antithrombin (AT), protein C and coagulation factors II through XII.

## 2. Materials and Methods

The experiment was approved by the institutional Ethics Committee at the Institute of Immunology and Experimental Therapy, Polish Academy of Sciences, Wroclaw, Poland (permission No. 63/2017). The animals were accommodated and cared for according to European Directive 2010/63/EU on the protection of animals used for scientific purposes. All procedures also adhered to the ARRIVE guidelines to make the results reliable and reproducible.

### 2.1. Animals

The study was carried out on 9 female Merino sheep ranging in age from 4 to 6 years and with a mean body weight of 48 kg (range 42–54). The sheep originated from a certified breeder (breed registration number PL049790823-001). The animals were clinically healthy and kept under normal husbandry conditions. No pharmacological treatment was administered for three weeks prior to the experiment. The health status of the animals before and during the study was checked by clinical examination and determination of total leukocyte count in peripheral blood (Table 1). During the experiment, sheep were fed ad libitum and had free access to water. Another 21 female sheep of the same breed, at a similar age and with a similar body weight, were also used for determining reference values of clotting parameters. The latter collection of animals constituted a non-surgical control group.

### 2.2. BM-MSC Isolation and Stimulation with FGF-2 and BMP-2

The process of BM-MSC preparation was carried out at the Laboratory of Biology and Neoplastic Cells, Hirszweld Institute of Immunology and Experimental Therapy, Polish Academy of Sciences, Wroclaw, Poland. A detailed description of the isolation and stimulation of BM-MSCs was described in a study by Gromolak et al. [24]. Briefly, four weeks before implantation of scaffolds, a portion of the bone marrow was aspirated from the iliac crest of the analgised experimental sheep. Within 30 min of bone marrow collection, BM-MSCs were harvested by centrifugation using a Lymphoflot (Bio-Rad, Dreieich, Germany) density gradient. The cells were washed with PBS and suspended in minimum essential medium α-transformation (αMEM, HIIET PAS, Wroclaw, Poland) supplemented with fetal bovine serum (Biowest, Riverside, MO, USA), L-glutamine and antibiotics, as well as 20 ng/mL of fibroblast growth factor-2 (FGF-2, Merck, Kenilworth, NJ, USA) and 100 ng/mL of bone morphogenetic protein-2 (BMP-2, Stem Cell Technologies, Saint-Egrève, France). Finally, 2 ml of autologous serum containing 20 million stimulated BM-MSCs was prepared for each experimental sheep.

### 2.3. Scaffold Fabrication and Preparation

The scaffolds used in the experiment were produced at the Faculty of Materials Science and Engineering, Warsaw University of Technology, Poland. They consisted of poly(ε-caprolactone) (PCL, PURASORB PC12, Corbion, Amsterdam, The Netherlands), hydroxyapatite (HAp, Ca5(OH)(PO4)3, particle size 33 nm, Merck, Feldkirchen, Germany) and tricalcium phosphate (β-TCP, Ca3(PO4)2, powder <4 μm, Progentix, Amsterdam, The Netherlands) with a mass ratio of 9:4.5:4.5. The three-dimensional porous structure of the scaffold was plotted using the BioScaffolder system (SysEng, Salzgitter, Germany) and formed by computer cutting to required dimensions of 20 mm × 20 mm × 12 mm (height × width × depth) (Figure 1a). Scaffolds in the final form were sterilized with a dose of 25 kGy of cobalt 60 gamma radiation and forwarded to the Department of Surgery, Faculty of Veterinary Medicine, Wroclaw University of Environmental and Life Sciences. Just before implantation, scaffolds for the particular sheep were washed with 2 mL of autologous serum containing 2 × 10^7^ BM-MSCs stimulated with FGF-2 and BMP-2. The effectiveness of scaffold settlement was assessed by fluorescence staining with 4′,6-diamidino-2-phenylindole (DAPI) (Figure 1b).

### 2.4. Surgical Procedure for Scaffold Implantation (SIP)

The animals were premedicated by intramuscular injection of 0.01 mg/kg of medetomidine (Cepetor^®^, CP-Pharma, Burgdorf, Germany), 1 mg/kg of butorphanol (Butomidor^®^, Richter Pharma AG, Wels, Austria) and 2 mg/kg of Zoletil (Zoletil^®^, Virbac, Carros, France). General anesthesia was induced with intravenous injection of propofol (Propofol-Lipuro 1%, B. Braun Melsungen AG, Melsungen, Germany) at an initial dose of 1 mg/kg and then at a dose based on the response of the animal. Sheep were intubated, and general anesthesia was maintained by oxygen-volatilized isoflurane inhalation (1.5–2.5 vol%, Iso-Vet, Piramal Healthcare, Morpeth, UK). Analgesia was performed with fentanyl (Fentanyl, WZF Polfa, Warsaw, Poland) in a continuous infusion (with initial dose of 2 mcg/kg and then 0.3 mcg/kg/min). The skin over the left mandibular area was shaved and scrubbed with 10% povidone iodine (Betadine 10%, Egis Pharmaceuticals Ltd., Budapest, Hungary). The mandible was exposed by the incision of the skin and muscles, and to prevent bleeding, the wound was electrocauterized (using the electrosurgical unit ERBE Erbotom, Tübingen, Germany). A bone fragment of the mandibular shaft measuring 2 by 2 cm was excised using a surgical oscillating saw (Stryker, Portage, MI, USA), accompanied by a copious irrigation with sterile saline. The scaffolds were introduced into the washed and dried bone defect. To fix the scaffold in place and to restore continuity of the mandible, an 8 by 1 cm titanium reconstruction bone plate (Medgal, Księżyno, Poland) was attached with four screws (Figure 2). The wound was closed by three layers of stitches—two layers of dissolving stiches for muscles and subcutaneous tissues (2-0, Monosyn, Braun, Melsungen, Germany) and one of non-resorptive suture for the skin (2-0, Novafil, Braun, Melsungen, Germany). After surgery, the sheep were placed in the housing units for postoperative care, which was performed by highly skilled animal caretakers. All animals were given 0.25 mg/kg of morphine intramuscularly (Morphini Sulfas, WZF Polfa, Warsaw, Poland) immediately after surgery and every 4 h for the two subsequent days: 0.2 mg/kg of meloxicam subcutaneously (Metacam, Boehringer Ingelheim, Ingelheim, Germany) and 30 mg/kg of metamizole intravenously (Pyralgivet, Vet-Agro, Lublin, Poland) once a day for 3 days after surgery.

### 2.5. Assessment of Hemostatic Parameters

Changes in parameters of general hemostasis were evaluated for 1 week after implantation of scaffolds. From each sheep, blood was collected at five defined time points—1 h before SIP (T0) and at the following times after SIP: 1 h (T1), 24 h (T2), 3 days (T3) and 7 days (T4). The determination of clotting parameters at T0 (before surgery) was intended to provide an additional control for the sheep that were then subject to scaffold implantation. In order to establish reference intervals and control data for the parameters investigated, blood was also taken from 21 healthy female sheep of the same breed at a similar age and with a similar body weight. The procedures of blood collection, obtaining plasma, as well as preparing of reagents and carrying out of all measurements, were accomplished according to the recommendations of the International Society on Thrombosis and Hemostasis (ISTH).

Blood was drawn from the jugular vein by a well-trained person using disposable 21 G needles. Because blood was also used for other tests, the specimens for the assessment of hemostatic parameters were collected second in the draw order. These samples were placed into 10-mL plastic tubes containing 3.8% sodium citrate (citrate:blood 1:9 *v*/*v*), mixed gently and forwarded to the laboratory within 2 h of collection. Plasma, obtained by centrifugation at 2500× *g*, was frozen at −20 °C for 1 week. Prior to analysis, plasma samples were thawed in a 37 °C water bath for 5 min and inspected visually for possible adverse reactions. Coagulated or hemolyzed samples were rejected.

All hemostatic parameters were analyzed in duplicate using the Coag Chrom 4000 coagulometer (Bio-Ksel, Grudziądz, Poland). Owing to the lack of sheep calibration plasma and corresponding materials dedicated to that animal species, human reagents and factor-deficient plasmas were used in our study to determine activity of hemostatic parameters. All the reagents required were reconstituted according to the manufacturer’s instruction (Bio-Ksel, Grudziądz, Poland).

The following hemostatic parameters were addressed in the present study: clotting times (PT and aPTT), concentrations of fibrinogen and D-dimer, as well as activities of antithrombin, protein C and clotting factors II-XII. All the procedures were carried out as described in our previous work [25]. Briefly, to determine PT and aPTT, 50 μL of plasma was mixed with 100 μL of preheated recombinant rabbit brain thromboplastin (for PT) or with 50 μL of a reagent containing colloidal silicon and synthetic phospholipids and with 50 μL of preheated calcium chloride (for aPTT). The time (in seconds) elapsed from mixing plasma to clot formation was measured by the coagulometer. The PT calibration curve was also used for calculating fibrinogen concentration. Antithrombin (AT) activity assay was carried out on 50 μL of diluted plasma mixed with 50 μL of a reagent containing thrombin (10 U/mL) and heparin. After adding a chromogenic substrate, the concentration of p-nitroaniline released was measured by the apparatus and converted into AT activity. Similarly, for protein C assessment, 50 μL of diluted plasma was mixed with 50 μL of protein C activator and then incubated with a chromogenic substrate. The concentration of p-nitroaniline was converted into protein C activity. The concentration of D-dimer was measured by an immunoturbidometric assay. A total of 90 μL of buffered plasma was preheated at 37 °C and then mixed with 90 μL of latex particles coated by monoclonal antibodies specific to D-dimer. The cloudiness of this mixture was measured by the coagulometer and converted into the amount of D-dimer. Activity assays for coagulation factors involved in both the extrinsic and common pathways (II, V, VII and X), as well as those involved in both the intrinsic and common pathways (VIII, IX, XI and XII), were performed by a modified one-stage PT or aPTT technique, respectively. An amount of 50 μL of diluted plasma was mixed with 50 μL of a corresponding human factor-deficient plasma and then with 100 μL of a reagent containing recombinant rabbit brain thromboplastin (for the first group of factors) or with 50 μL of aPTT reagent containing colloidal silicon and synthetic phospholipids and with 50 μL of a corresponding human factor-deficient plasma, followed by adding 50 μL of preheated calcium chloride (for the second group). The time (in seconds) elapsed from mixing of test plasma and factor-deficient plasma with a reagent to clot formation was measured by the coagulometer, and results were expressed as percentage activity of a given factor in the test plasma in relation to that of the deficient plasma.

### 2.6. Statistical Analysis

Statistical analysis was performed using STATISTICA 13 software (StatSoft Inc., Tulsa, OK, USA). Data were analyzed for normal distribution by the Shapiro-Wilk test. The one-way analysis of variance (ANOVA) was performed. Obtained results are presented as means and standard deviation. Dunnett’s test was used to determine differences between means. Differences at *p* < 0.01 and *p* < 0.05 levels were considered statistically significant.

## 3. Results

Leukocyte counts in the experimental animals are shown in Table 1. Surgical intervention resulted in a decrease in this parameter (to 3.5 × 10^9^/L at T1). In the following few days, it rose markedly (with the maximum peak of 7.9 × 10^9^/L at T2) and went back to the basic value at T4.

**Table 1 materials-14-06934-t001:** Total leukocyte counts in the experimental sheep. Data are expressed as means and ± standard deviations (SD).

Assay		1 h before SIP (T 0)	1 h after SIP (T 1)	24 h after SIP (T 2)	3 Days after SIP (T 3)	7 Days after SIP (T 4)	Reference Range ^‡^
Leukocyte count (×10^9^/L)	mean	5.4	3.5	7.9	7.2	5.9	3.6–9.5
	SD	±1.2	±0.9	±1.1	±1.2	±1.3	

SIP—scaffold implantation procedure; ^‡^ based on results from 21 healthy sheep.

Hemostatic parameters measured within this experiment are presented in Table 2 (PT, aPTT) and Table 3 (clotting factors II through XII, AT, protein C), as well as in Figure 3a (fibrinogen concentration) and Figure 3b (D-dimer concentration).

The surgical procedure of scaffold implantation resulted in slight prolongation of both PT and aPTT (Table 2). This phenomenon was observed as early as 1 h after surgery and continued for the whole observation period. Prolongation of PT, although it did not exceed one second at T1, was statistically significant at *p* < 0.05 when compared with PT values measured before implantation (T0). The longest PT (14.8 s) exceeding the upper reference limit was detected one day after surgery (T2). This difference amounted to 2.4 s (19.4%) and was significant at *p* < 0.01. On the subsequent days (T3 and T4), even if PT was shorter then at T2 and lay within reference ranges, it remained longer than at T0 (on day 7, this value was longer by 1 s and differed at *p* < 0.05 from that at T0). A similar tendency was observed for the dynamics of aPTT. This parameter was longest at T2 (33.9 s), exceeding the upper reference limit and outstripping that at T0 (26.5 s) by 7.4 s (~28%, statistically significant at *p* < 0.01 level). Next, although aPTT values returned to the reference range (amounting to 27.0 s and 30.3 s at T3 and T4, respectively), they were longer than those detected prior to surgery (T0). The difference between aPTT at T4 and T0 was significant at *p* < 0.05.

Activities of clotting factors (clotting activators) varied in response to the implantation procedure (Table 3). At times immediately after surgery (T1 and T2), a decreased activity of most factors (i.e., II, VII, VIII, IX, X and XII) was observed. In comparison to T0, these values were statistically significant for factor II (at T1, *p* < 0.01) and IX (at T2, *p* < 0.05). On the contrary, factor XI displayed an increased activity at T1, which fell at T2 to values similar to those measured before implantation. In the next days (T3 and T4), all the factors investigated raised their activities (as compared to those at T2), with statistically significant values detected for factor IX (at T3, *p* < 0.05) and factor XI (at T3, *p* < 0.01). Taking the time just before implantation (T0) as a baseline, these activities measured at T3 and T4 showed a marked variability. They were higher than those at T0 (and exceeded the upper reference limit) in the case of factor V (difference at *p* < 0.01 at both T3 and T4), as well as factors X and XI (the latter being different at *p* < 0.01 at T3). For factors II and VII, activities were also increased in relation to those at T0 but remained within the reference ranges (the first of them displayed a significant difference at *p* < 0.01 at T3 and T4). In turn, the values measured at T3 and T4 for both factors VIII and IX were lower than at T0.

Levels of both clotting inhibitors (AT and protein C; Table 3) were decreased in the period just after surgery (except protein C level at T1, which was practically unchanged). Lower activity of AT was measured as early as one hour after implantation of the scaffold (T1, *p* < 0.01), continued for the next day (T2, *p* < 0.05) and went back to the pre-surgery values on day 7 (T4). Protein C activity diminished somewhat later, only 24 h after implantation, and for a short time. From day 3 onwards, it rose continuously, exceeding, at that time, the T0 level. At the end of the measurement period (T4), protein C activity was even higher, and although still remaining within the reference range, was statistically different at *p* < 0.01.

## 4. Discussion

The development of various types of scaffolds has become competitive and to be seen as an effective replacement for the previous gold standard in tissue engineering, i.e., autologous grafting [1,2,5,7,19]. A major advancement in this field has also been achieved through the settlement of a scaffold with BM-MSCs stimulated by or incorporated with various growth factors, especially FGF-2 and BMP-2 [1,7,24]. These procedures result in faster and better regeneration of bones, soft tissues and blood vessels, and consequently, lead to the shortening of the healing process, prevent adverse events of hypoxia and inflammation, help avoid apoptosis and protect from host immune attack [1,2,6]. However, some materials and growth factors used in composite scaffolds have been introduced relatively recently, and a number of possible host reactions have not been studied thoroughly so far. For example, there is a lack of literature data regarding in vivo hemostasis phenomena in sheep subject to procedures of scaffold implantation.

The healing process of bone fractures, bone–soft tissue defects or surgical reconstructive interventions using substitute materials consists of four stages: (a) inflammation (up to one week after injury), (b) formation of soft callus, (c) formation of hard callus and (d) bone remodeling [13,14,26,27]. At the beginning of the first stage, following destruction of blood vessels and the contact of scaffold material with the patient’s blood, there is a formation of hematoma and clot at the site of implantation [4,14,16,18]. Extravasation activates mechanisms of hemostasis, leading to the formation of a clot. The fibrin-fixed blood clot constitutes an additional mechanical support to the implanted scaffold and closes the wound, preventing further blood loss. Thrombin and thrombin cleavage products act as chemotactic signals, whereas fibrin forms a natural framework for the inflowing cells that are involved in the process of removal of damaged tissues, debris and bacteria and participate in regenerative/reparative phenomena [16,17,20,26]. The aggregated and activated platelets release substances involved in the processes of coagulation and fibrinolysis, inflammatory mediators, immune-modulating factors and some antimicrobial substances, as well as osteogenic and angiogenic factors [4,12,15,17]. At a later time, however, in order to make the healing processes more effective (i.e., for the formation of soft- and hard-tissue callus, vascularization and proper bone repair), the hemostatic clog is dissolved by the protease plasmin (in the process known as fibrinolysis). Plasmin is additionally considered to have the abilities of decreasing the inflammatory process, preventing calcium deposition in soft tissues (heterotopic ossification) and curbing undue callus formation [18].

The above phenomena rely on the sequential activation of particular components of hemostasis. In the literature, there are a number of reports describing the involvement of platelets (primary hemostasis) and plasma proteins (secondary hemostasis and fibrinolysis) in the regenerative processes at the site of implantation. Most of those studies have been focused on mechanisms of local hemostasis and concern qualitative changes only [4,17,18]. Little is known, however, about the mechanisms of general hemostasis, which could also have an impact on the outcome of implantation. Destruction of tissues following surgery and the introduction of a foreign body (scaffold), which is often additionally settled with specific cells and impregnated by biomodulating substances, not only infringe on the integrity of the host but can also affect hemostatic and immune mechanisms [3,6,8].

In the light of the above, we focused on the reactivity of general hemostasis in the ovine model in response to the introduction of a composite scaffold (HAp, β-TCP and PCL) settled with bone marrow-derived mesenchymal stem cells stimulated by fibroblast growth factor-2 and bone morphogenetic protein-2. We used a 7-day observation period in our experiment, corresponding to the first phase of post-operative wound healing. Measurements of a broad spectrum of hemostatic parameters were carried out one hour and 1, 3 and 7 days after scaffold implantation.

Just one hour after the surgical procedure (T1), we observed changes in all the clotting parameters examined, except protein C and D-dimer. Simultaneous prolongation (to a small degree) of both aPTT and PT at this stage of the experiment suggests a shift of hemostatic balance towards a slowdown of clotting mechanisms (hypocoagulable state). This was accompanied by an initial decrease in activities of almost all clotting factors (except XI), a marked decrease in AT activity (into 61.9%) and a reduction in fibrinogen level, probably resulting from their loss and consumption following controlled surgery bleeding. That conjecture can be confirmed by the observed decrease in WBC count to 3.5 x 10^9^/L. The first day after surgery (T2) was a breakthrough moment in the hemostatic reaction to scaffold implantation in the experimental sheep. There was a further prolongation of aPTT and PT (into 33.9 and 14.8 s, respectively), indicating an even more pronounced slowing of hemostasis. However, these parameters only slightly exceeded the upper reference limits (by 1.4 and 0.3 s), and the excessive reaction may have been prevented by mediators of clotting, which could be released from activated platelets, produced as a part of the inflammatory reaction or activated from precursors circulating in blood [12,17,28,29]. At this time, the sheep subjected to surgery displayed an increased activity of clotting factors II, V, X, and AT, as well as an increase in fibrinogen concentration. Leukocytosis (WBC 7.9 × 10^9^/L, as compared to the initial count of 5.4 × 10^9^/L) and the decrease in protein C activity (the negative acute-phase protein) [28] were evidence of commencing inflammation [30,31]. The notion of a mild compensation of clotting slowdown can be supported by the lowered (than at T1) activities of factors VIII, IX, XI and XII, indicating that the apparent activation of the amplification phase of coagulation (intrinsic pathway), while subsequent fibrinolysis was not yet initiated [29,30]. The observed increase in D-dimer concentration (312 μg/L) was slight and remained near the upper reference limit for this parameter (300 μg/L). Additionally, D-dimer levels differed minimally from those before implantation (279.9 μg/L). The activity of clotting factor VII continued to be low (24.6%), suggesting that tissues and cells were no longer damaged.

In the following days, the process of counteracting the slowdown clotting mechanisms was even more pronounced. There was a shortening of PT and aPTT, as well as an increase (to a different degree) in activities of all modulators of hemostasis. On day 3 (T3), a significant rise in activities of factors II through XII was detected (in comparison to the results obtained on the first day after surgery). The values measured for factors II, V, VII and X were also higher than those before implantation. The activities of factors V, X and XI exceeded even the respective upper reference limits. Clotting inhibitors (AT and protein C), as well as fibrinogen, also increased their concentrations at T3. Fibrinolysis was, however, not observed at that time. The concentration of D-dimer decreased to the values observed at T0 and T1. The WBC count (7.2 × 10^9^/L) remained at a level comparable to that measured one day after implantation but was distinctly higher than before surgery. The changes described, i.e., the increase in activities of all modulators of hemostasis (activators and inhibitors), as well as in concentration of fibrinogen, are evidence of strengthening and acceleration of the clotting process, which, in turn, is associated with the ongoing inflammatory reaction and launching of the amplification phase of coagulation (intrinsic pathway), acting as a link between inflammation and hemostasis [28,29]. On day 7 (T4), although the overall tendency in changes of the hemostatic parameters described above was roughly the same (when compared with those at the 24th hour after surgery), some differences compared to the results obtained previously (at T0 and T3) were noticed. In comparison to T3, there was a slight prolongation of both times measured (PT and aPTT). The activities of factors II, VIII, X and XII, as well as AT and protein C, were still on the rise. The same applied to the concentrations of fibrinogen and D-dimer. However, the dynamics of the abovementioned increases in clotting parameters were weaker. The probable slowdown of acute inflammation (as deduced from the lower WBC count) contributed to a lesser synthesis of the corresponding proteins [17,29]. In contrast, lower activities of factors V, VII, IX and XI, accompanied by the decreased WBC count (amounting to 5.9 × 10^9^/L – only just above the number measured before surgery), were also detected. These lower activities of the remaining factors could, in part, be attributable to the action of clotting inhibitors (e.g., factor V is inactivated by protein C), a more rapid depletion of these factors or launching of fibrinolysis (breakdown of some factors by plasmin) [28,31].

When analyzing the mean results of the clotting parameters in sheep, one can notice a time shift between the change in activities of factors regulating hemostasis processes and the moment of manifestation of the corresponding effects. For example, alterations in activities of activators (clotting factors II through XII) and inhibitors (AT and protein C) occurred at least one day in advance of the appearance of changes in hemostasis (prolongation or shortening of PT and aPTT) and fibrinolysis (fluctuations in D-dimer concentrations). The phenomenon of time inconsistency was also observed in relation to some aspects of the cooperation of hemostasis and inflammation. Following scaffold implantation (at T2), there was a rapid increase in the acute-phase protein fibrinogen; however, activation of the intrinsic pathway (a link between the two above processes) took place only after two days (at T3).

Unfortunately, owing to the lack of the corresponding literature data, it is very difficult to corroborate the above-described phenomena. For the same reason, the overall comparison of the results pertaining to general hemostasis in individuals subject to scaffold implantation is nearly impossible. The reports available focus mainly on in vitro studies, particularly those analyzing reactions following the contact of plasma with hydroxyapatite (HAp) and its composites and refer to limited clotting parameters only. For example, Arimura et al. [19] exposed human plasma to HAp (as a pure powder or mixture with agarose gel) for a short time (1 min for determining PT and 2 min for aPTT). They observed a prolongation of aPTT in the case of unmixed HAp (anticoagulant activity) or shortening of this parameter when the mixture was used (procoagulant activity). In no case, however, was PT affected. As for the reason for the observed changes in aPTT, the authors assumed a diverse action of HAp crystals, varying in size, on fibrinogen and other clotting factors. In a similar experiment, Santos et al. [32] determined PT and aPTT after a 3-h exposure of two types of HAp to platelet-poor plasma. The authors detected only minor (not exceeding a few seconds) differences in the parameters measured that depended on the amount of HAp in the specimen used. Yet another group of investigators [33] examined the impact of various materials, including HAp, on in vitro clotting processes in human blood. HAp was shown to be a recommended hemostatic agent for bone engineering. However, the results obtained in that experiment were expressed as a blood-clotting index (calculated from spectrophotometric measurements of blood contacted with HAp and blood mixed with citrate dextrose) and thus could not be compared with our results. Furthermore, in all the above studies, different experimental conditions and scaffold materials were used.

The only report available that describes results of coagulation tests measured in an in vivo experiment is that performed on mice and rats implanted with a biodegradable polymer scaffold [34]. The authors determined PT and aPTT only (however, the time point of blood collection was not specified) and found no significant differences between the experimental and control groups. The use of different animals and study conditions, however, also preclude reliable inference.

## 5. Conclusions

In conclusion, the mean values of most of the clotting parameters in sheep measured at particular times before and after implantation differed only a little and lay within specific reference ranges. In the case of a few parameters only, we detected increases in activity or concentration of the values exceeding the upper limit. The latter applied to some clotting factors and fibrinogen. As for factor X, such differences in activities at day 3 and day 7 after implantation were only subtle and amounted to 14.5% and 17%, respectively. A much larger shift was observed for factors V and XI, which increased their activities by 46% and 170%, respectively, at day 3. However, this condition continued for a short time only and started to quickly recede. The concentration of fibrinogen soared at day 3 (by 85%) and was still on the rise (though at a slower rate) at day 7. All these changes, however, had no impact on the efficiency and control capabilities of hemostatic processes in the experimental animals. Differences in clotting times (PT and aPTT) measured throughout the observation period were so subtle that they may be considered a tendency rather than true changes in general hemostasis. In fact, they were more related to a mild inflammatory condition resulting from the impairment of tissue integrity following surgery, which is inevitable in therapeutic bone-soft tissue engineering. Thus, taking into account results of our experiment, a conclusion can be reached that both the use of a composite scaffold settled with BMP-2- and FGF-2-stimulated mesenchymal stem cells and the implantation technique applied were safe and did not incur the risk of thromboembolic or bleeding events in the experimental sheep.

## Figures and Tables

**Figure 1 materials-14-06934-f001:**
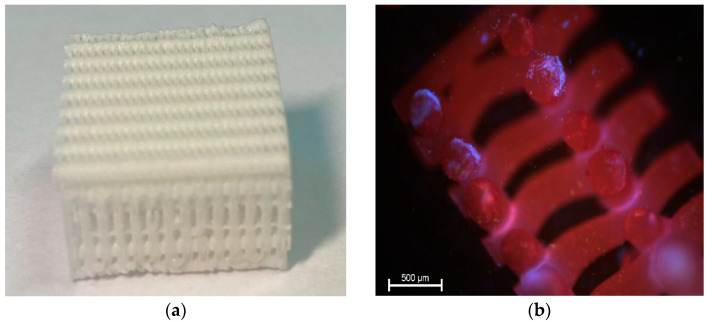
Photograph of the 3D printed composite scaffold used in the present study (**a**). Fragment of the scaffold settled with BM-MSCs (shown in blue) stained with DAPI. Magnification ×40 (**b**).

**Figure 2 materials-14-06934-f002:**
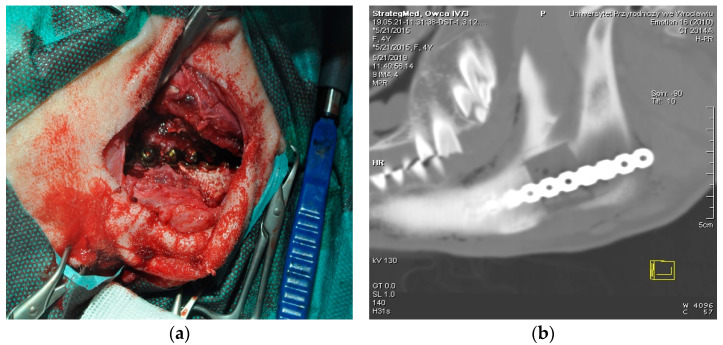
Procedure of scaffold implantation. (**a**) Surgical wound showing titanium bone plate attached with srews. (**b**) CT image of the scaffold placed in the ovine mandible.

**Figure 3 materials-14-06934-f003:**
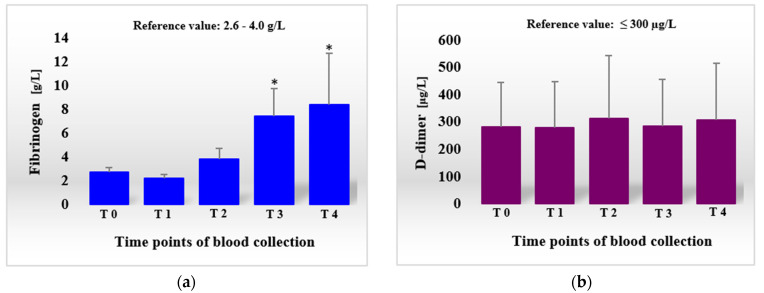
Concentrations of fibrinogen (**a**) and D-dimer (**b**) in the sheep before and after scaffold implantation procedure (SIP). Data are expressed as means and ± standard deviations. T0, 1 h before SIP; T1, 1 h after SIP; T2, 24 h after SIP; T3, 3 days after SIP; T4, 7 days after SIP.; * difference statistically significant at *p* ˂ 0.01, as compared with T0.

**Table 2 materials-14-06934-t002:** Measurements of PT and aPTT in the experimental sheep. Data are expressed as means and ± standard deviations (SD).

Assay		1 h before SIP (T 0)	1 h after SIP (T 1)	24 h after SIP (T 2)	3 Days after SIP (T 3)	7 Days after SIP (T 4)	Reference Range ^‡^
PT (s)	mean	12.4	13.3 *	14.8 **	13.0	13.4 *	9.7–14.5
	SD	±1.2	±0.9	±1.1	±1.2	±1.3	
aPTT (s)	mean	26.5	27.1	33.9 **	27.0	30.2 *	21.7–32.5
	SD	±5.1	±3.9	±3.3	±6.5	±4.3	

PT, prothrombin time; aPTT, activated partial thromboplastin time; SIP, scaffold implantation procedure; ^‡^ based on results from 21 healthy sheep.; * differences statistically significant at *p* ˂ 0.05, as compared to T0.; ** differences statistically significant at *p* ˂ 0.01, as compared to T0.

**Table 3 materials-14-06934-t003:** Coagulation activators (factors II–XII) and inhibitors (AT and protein C) in the experimental sheep. Data are expressed as means and ± standard deviations (SD).

Assay		1 h before SIP (T 0)	1 h after SIP (T 1)	24 h after SIP (T 2)	3 Days after SIP (T 3)	7 Days after SIP (T 4)	Reference Range ^‡^
Factor II (% activity)	mean	27.6	21.1 ^a^	23.8	35.1 ^a^	36.0 ^a^	27.0–40.6
	SD	±2.8	±2.4	±3.3	±4.3	±4.9	
Factor V (% activity)	mean	294.2	255.7	310.3	515.3 ^a^	443.6 ^a^	234.6–352.0
	SD	±43.1	±43.7	±50.8	±132.4	±109.1	
Factor VII (% activity)	mean	36.4	26.6	24.6	53.5	49.2	36.9–55.3
	SD	±25.0	±14.7	±15.9	±28.2	±25.8	
Factor VIII (% activity)	mean	606.7	574.8	468.7	554.8	579.1	507.0–760.4
	SD	±157.3	±162.4	±190.4	±183.7	±170.4	
Factor IX (% activity)	mean	512.3	297.5	266.2 *	489.7 **	448.3	439.2–658.8
	SD	±298.1	±242.1	±288.1	±238.9	±334.9	
Factor X (% activity)	mean	50.0	36.8	37.6	58.5	59.8	34.1–51.1
	SD	±19.3	±15.1	±14.5	±21.5	±21.7	
Factor XI (% activity)	mean	37.6	46.9	35.2	187.3 ^a,b^	77.5 ^c^	46.3–69.4
	SD	±99.2	±83.9	±124.0	±245.4	±140.9	
Factor XII (% activity)	mean	73.5	38.3	28.6	52.9	76.3	109.8–164.8
	SD	±52.4	±25.9	±13.1	±27.7	±57.0	
AT (% activity)	mean	78.0	61.9 ^a^	67.3 *	73.6	76.4	66.7–100.1
	SD	±11.6	±12.5	±13.1	±13.6	±9.9	
Protein C (% activity)	mean	48.9	50.0	42.2	57.0	62.6 ^a^	46.2–69.2
	SD	±7.7	±8.4	±8.5	±14.7	±13.8	

AT, antithrombin; SIP, scaffold implantation procedure; ^‡^ based on results from 21 healthy sheep.; differences statistically significant at *p* ˂ 0.05, as compared to: * T0, ** T2. Differences statistically significant at *p* ˂ 0.01 as compared to: ^a^ T0, ^b^ T2, ^c^ T3.

## Data Availability

The data presented in this paper is available with request from the corresponding author.

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
