# Peer review of "Parameters of Hemostasis in Sheep Implanted with Composite Scaffold Settled by Stimulated Mesenchymal Stem Cells—Evaluation of the Animal Model"

_materials, 2021, doi:10.3390/ma14226934_

Round 1

Reviewer 1 Report

This is very unique study. Well written, although this looks long and a little complicated for me. However, all scaffolds contained growth factors and mesenchymal stem cells. This was conducted as a single arm study, and did not have any control groups. I’d like you to the authors discuss the application of scaffolds without these factors.

Followings were minor revisions.

  • Generally, bone morphogenetic protein-2 is described as BMP-2, but here, described as MBP-2. Please reconsider.
  • Table 1 may be one of results. It is better to describe in Results.
  • In Materials and Methods, the description of medium might not be correct. Please check.
  • In Table 2, the number of line (258) was contained in Table.

Author Response

1/ All changes to the former version (deletions and insertions) were made using the “Track
Changes” function.

2/ We added 4 pictures showing the scaffold used in our experiment, scaffold settled with mesenchymal stem cells and surgical procedure of scaffold implantation.

This is very unique study. Well written, although this looks long and a little complicated for me. However, all scaffolds contained growth factors and mesenchymal stem cells. This was conducted as a single arm study, and did not have any control groups. I’d like you to the authors discuss the application of scaffolds without these factors.

Followings were minor revisions.

  • Generally, bone morphogenetic protein-2 is described as BMP-2, but here, described as MBP-2. Please reconsider.
  • Table 1 may be one of results. It is better to describe in Results.
  • In Materials and Methods, the description of medium might not be correct. Please check.
  • In Table 2, the number of line (258) was contained in Table.

Response:

Our experiment was intended to assess hemostatic phenomena in sheep subject to implantation of scaffolds washed with stimulated mesenchymal stem cells as a complete procedure. We realize that there would be better to use and analyze individual factors (that is, surgery without scaffolds and implanting scaffolds not settled with BM-MSCs), however, the restricted conditions imposed by the Local Ethics Committee (and limited number of animals that could be used for the experiment) precluded the employment of so many statistically reliable groups of animals. The control animals that we used were 21 healthy sheep for establishing reference values of hemostatic parameters (i.e. non-surgical animals; the consent of the Committee was not necessary as blood was collected during routine health check of this animals). Nine sheep were experimental animals (clotting parameters estimated at T0, that is, before surgery, were considered as an additional control for this group).

Minor comments:

1/ The abbreviation “BMP-2” is used throughout the text.

2/ Table 1 was moved to the Results section.

3/ The name of medium used was corrected (the mistake was probably the result of conversion to the journal format)

4/ Table 2 was corrected

Reviewer 2 Report

The manuscript is well written and the concept of the research is appreciable. I have few suggestions to improve the overall presentation of the paper.

  1. Abstract is little bit boring because no numerical value is given while presenting the results.
  2. Introduction is too long reduce it to one and half page.
  3. Heading of the Table is not align.
  4. Please add the conclusion part to the article

Author Response

1/ All changes to the former version (deletions and insertions) were made using the “Track Changes” function.

2/ We added 4 pictures showing the scaffold used in our experiment, scaffold settled with mesenchymal stem cells and surgical procedure of scaffold implantation.

  1. Abstract is little bit boring because no numerical value is given while presenting the results.
  2. Introduction is too long reduce it to one and half page.
  3. Heading of the Table is not align.
  4. Please add the conclusion part to the article

Response:

1/ Some numerical values (clotting times and the concentration of fibrinogen) were added to Abstract.

2/ Introduction was shortened.

3/Table headings were corrected.

4/ We added the Conlusions section to the manuscript.

Reviewer 3 Report

In this manuscript, the authors aimed to investigate changes in hemostatic parameters in sheep implanted with a scaffold. Multiple hemostatic parameters were assessed at a series of time points, including 1 hour before and 1 hour, 24 hours, 3 days and 7 days after scaffold implantation. The manuscript was overall well-written and involved an interesting and clinically relevant topic, however, there were several flaws in experimental design, and data presentation.

The biggest drawback is the lack of proper controls. All the 21 sheep received the same treatment, all the hemostatic parameters at postsurgical time points were compared to those parameters at T0. The study would be more scientifically sound and significant if there were other control groups, for example, non-surgical groups, surgical (trauma) without scaffold implantation, surgical + scaffold but without MSC or without BMP2/FGF-2 treatment. Without proper controls, the results shown here could be due to the trauma from the surgery and subsequent natural healing,

More data on the scaffold should be provided. Did the MSC adhere to the scaffold successfully? Any SEM images for the scaffold or staining for the MSC? Without proving the MSC scaffold is successful, the conclusion of “both the use of a composite scaffold settled with BMP-2- and FGF-2-stimulated mesenchymal stem cells and the implantation technique applied were safe and did not incur the risk of thromboembolic or bleeding events in the experimental sheep (Page 11, Line477-480) doesn’t seem to be solid.

Author Response

1/ All changes to the former version (deletions and insertions) were made using the “Track Changes” function.

2/ We added 4 pictures showing the scaffold used in our experiment, scaffold settled with mesenchymal stem cells and surgical procedure of scaffold implantation.

The biggest drawback is the lack of proper controls. All the 21 sheep received the same treatment, all the hemostatic parameters at postsurgical time points were compared to those parameters at T0. The study would be more scientifically sound and significant if there were other control groups, for example, non-surgical groups, surgical (trauma) without scaffold implantation, surgical + scaffold but without MSC or without BMP2/FGF-2 treatment. Without proper controls, the results shown here could be due to the trauma from the surgery and subsequent natural healing,

More data on the scaffold should be provided. Did the MSC adhere to the scaffold successfully? Any SEM images for the scaffold or staining for the MSC? Without proving the MSC scaffold is successful, the conclusion of “both the use of a composite scaffold settled with BMP-2- and FGF-2-stimulated mesenchymal stem cells and the implantation technique applied were safe and did not incur the risk of thromboembolic or bleeding events in the experimental sheep (Page 11, Line477-480) doesn’t seem to be solid.

Response:

1/ Our experiment was intended to assess hemostatic phenomena in sheep subject to implantation of scaffolds washed with stimulated mesenchymal stem cells as a complete procedure. We realize that there would be better to use and analyse individual factors (that is, surgery without scaffolds and implanting scaffolds not settled with BM-MSCs), however, the restricted conditions imposed by the Local Ethics Committee (and limited number of animals that could be used for the experiment) precluded the employment of so many statistically reliable groups of animals. The control animals that we used were 21 healthy sheep for establishing reference values of hemostatic parameters (i.e. non-surgical animals; the consent of the Committee was not necessary as blood was collected during routine health check of this animals). Nine sheep were experimental animals (clotting parameters estimated at T0, that is, before surgery, were considered as an additional control for this group). We agree that changes in hemostatic parameters in the experimental animals may have resulted, to some degree, from the surgical trauma (such a statement is also present in the manuscript, L. 474-476 in the former version, 502-504 in the corrected one), however, the process of scaffold implantation is always associated with damage of tissues and both factors (trauma and the effect of implanting scaffold) could be jointly taken into consideration. All in all, our experiment showed that the material used, that is, a composite of poly(ε-caprolactone), hydroxyapatite and tricalcium phosphate, seems to affect only little (if any) hemostatic processes in the sheep.

2/ We have added a picture showing BM-MSCs attached to the composite scaffold (Figure 2b).

Round 2

Reviewer 3 Report

The authors have carefully revised the manuscript and addressed most of the reviewers' questions.

This manuscript is a resubmission of an earlier submission. The following is a list of the peer review reports and author responses from that submission.